# Multi-neuron intracellular recording in vivo via interacting autopatching robots

**Suhasa B Kodandaramaiah**[1,2,3,4†§], **Francisco J Flores**[5,6†], **Gregory L Holst**[4], **Annabelle C Singer**[1,2#], **Xue Han**[7], **Emery N Brown**[5,6,8], **Edward S Boyden**[1,2,3‡*], **Craig R Forest**[4‡*]

[1]Media Lab, Massachusetts Institute of Technology, Cambridge, United States; [2]McGovern Institute for Brain Research, Massachusetts Institute of Technology, Cambridge, United States; [3]Department of Biological Engineering, Massachusetts Institute of Technology, Cambridge, United States; [4]G.W. Woodruff School of Mechanical Engineering, Georgia Institute of Technology, Atlanta, United States; [5]Department of Anesthesia, Critical Care and Pain Medicine, Massachusetts General Hospital, Boston, United States; [6]Picower Institute for Memory and Learning, Massachusetts Institute of Technology, Cambridge, United States; [7]Department of Biomedical Engineering, Boston University, Boston, United States; [8]Institute for Medical Engineering and Science, Massachusetts Institute of Technology, Cambridge, United States

**\*For correspondence:**
esb@media.mit.edu (ESB);
cforest@gatech.edu (CRF)

[†]These authors contributed equally to this work
[‡]These authors also contributed equally to this work

**Present address:** [§]Department of Mechanical Engineering, University of Minnesota-Twin Cities, Minneapolis, United States; [#]Coulter Department of Biomedical Engineering, Georgia Institute of Technology and Emory University, Atlanta, United States

**Competing interests:** The authors declare that no competing interests exist.

**Abstract** The activities of groups of neurons in a circuit or brain region are important for neuronal computations that contribute to behaviors and disease states. Traditional extracellular recordings have been powerful and scalable, but much less is known about the intracellular processes that lead to spiking activity. We present a robotic system, the multipatcher, capable of automatically obtaining blind whole-cell patch clamp recordings from multiple neurons simultaneously. The multipatcher significantly extends automated patch clamping, or 'autopatching', to guide four interacting electrodes in a coordinated fashion, avoiding mechanical coupling in the brain. We demonstrate its performance in the cortex of anesthetized and awake mice. A multipatcher with four electrodes took an average of 10 min to obtain dual or triple recordings in 29% of trials in anesthetized mice, and in 18% of the trials in awake mice, thus illustrating practical yield and throughput to obtain multiple, simultaneous whole-cell recordings in vivo.
DOI: https://doi.org/10.7554/eLife.24656.001

## Introduction

Mammalian brains consist of neurons organized into densely interconnected circuits. Traditionally circuit-level characterization of neuronal activities has been carried out using extracellular recording probes (*Buzsáki, 2004*) or using genetically encoded calcium indicators (*Chen et al., 2013*). While these methods reveal supra-threshold spiking activities of individual neurons in a circuit, they cannot examine synaptic and subthreshold events in neurons, important for understanding the processes within and between cells that lead to spiking. While a few studies have performed simultaneous multi-neuron intracellular recordings in vivo (*van Welie et al., 2016*; *Jouhanneau et al., 2015*; *Poulet and Petersen, 2008*), the difficulty of the technique limits its use.

Recently we developed a robot, called the 'autopatcher', that performs fully-automated whole-cell patch clamping of single neurons in the living mouse brain (*Kodandaramaiah et al., 2012*; *Kodandaramaiah et al., 2016*). The autopatcher uses pipette impedance measurements to hunt for

neurons, followed by gigasealing and break-in using electronically controlled pressure regulators. We explored several straightforward attempts to scale up this approach to achieve multi-neuron recordings, but they exhibited very low yield. From these pilot studies, we hypothesized that robots engaged in stationary activities, such as the delicate task of gigasealing, would be disrupted by robots engaged in motion, such as during the task of neuron hunting. Therefore, an optimal combination of hardware and software should optimize the interaction between stationary and active tasks across pipettes.

We thus devised interactions between multiple autonomous patch robots, such that when one was attempting a stationary task, the others would wait before moving. Using this approach, we controlled a four electrode multipatcher robot and used it to perform multi-neuron recordings in the somatosensory and visual cortices of anesthetized mice, and the somatosensory cortex of the awake mouse. Under anesthetized conditions, the multipatcher robot obtained dual or triple whole-cell recordings in 29% of trials, and at least one whole-cell recording in 90% of trials. An individual robot had the same whole-cell yield when used in our multipatcher robot as it did when working alone (31%). In awake mice the multipatcher obtained dual or triple whole-cell recordings in 18% of the trials. The robot took 10.5 $\pm$2.6 min to complete each trial (summary statistics given as mean $\pm$ S.D. throughout the paper, unless otherwise noted), with recordings lasting 14.0 $\pm$10.0 min in the awake head-fixed mouse. Thus, the multipatcher is a practical solution for performing intracellular recordings from multiple neurons in the intact brain.

## Multipatcher hardware

The multipatcher robot was built by extending some of the hardware components of our previously described autopatcher robot (*Kodandaramaiah et al., 2012*). It consists of an array of four robotic arms for manipulating the patch recording pipettes, patch amplifiers, a signal control box, a computer with an interface board, and a pressure control box (*Figure 1*, *Figure 1—figure supplements 1* and *2*, and *Supplementary files 1*, *2* and *3*). The pipette arms' actuators and motors allowed programmatic control of the pipettes' positions during robot operation. The four arms were arranged in a radial array (*Figure 2c*) to enable tips of patch pipettes to be positioned and manipulated within 50 $\mu$m of each other inside the brain. The internal pressures of the patch pipettes were independently controlled using the pressure control box (*Figure 1—figure supplement 1*).

### Robotic arms

The primary components of a robotic arm include the following: patch pipette, pipette holder, pipette holder extension, and amplifier headstage. The headstage is mounted onto on a programmable linear motor using a custom dovetail adapter plate (PT1-Z8 motor with TDC001 controller, Thorlabs) allowing controlled actuation of the pipette during robot operation (*Figure 2a and b*). This programmable linear motor was mounted at an angle of 60° (relative to the horizontal plane, *Figure 2a*) using a swivel mount adapter (FG-285210, Sutter Instruments) on a manually controlled three-axis manipulator (MPC285, Sutter Instruments) to position the pipette outside the brain. Four such robotic arms were arranged with rotational symmetry (*Figure 2c and d*). This arrangement, combined with the ability to precisely open arrays of craniotomies (*Pak et al., 2015*) allowed the tips of patch pipettes to be positioned and manipulated very close to each other (*Figure 2d*, inset), as close as 50 $\mu$m between tips when positioned on the brain's surface.

### Signal control box

Signals from the amplifier headstages are sent to two dual-channel patch amplifiers (Multiclamp 700B, Molecular Devices). Amplified signals were digitized in two computer interface boards: a computer interface board present inside the signal control box, and an external computer interface board, or digitizer (*Figure 1—figure supplement 1*). The computer interface board inside the signal control box (cDAQ-9174 chassis with modules NI 9215 for analog inputs, NI 9264 for analog outputs and NI 9375 for digital outputs, National Instruments) was utilized for multipatcher operation and the digitizer (Digidata 1440B, Molecular Devices) was dedicated to data-acquisition after whole-cell patch clamp recordings were obtained. Command signals to each patch amplifier were sent from analog output channels on either the cDAQ-9174 (during the multipatcher operation) or from the digitizer (during whole-cell recording). BNC signaling relays (CX230, Tohtsu) route the command

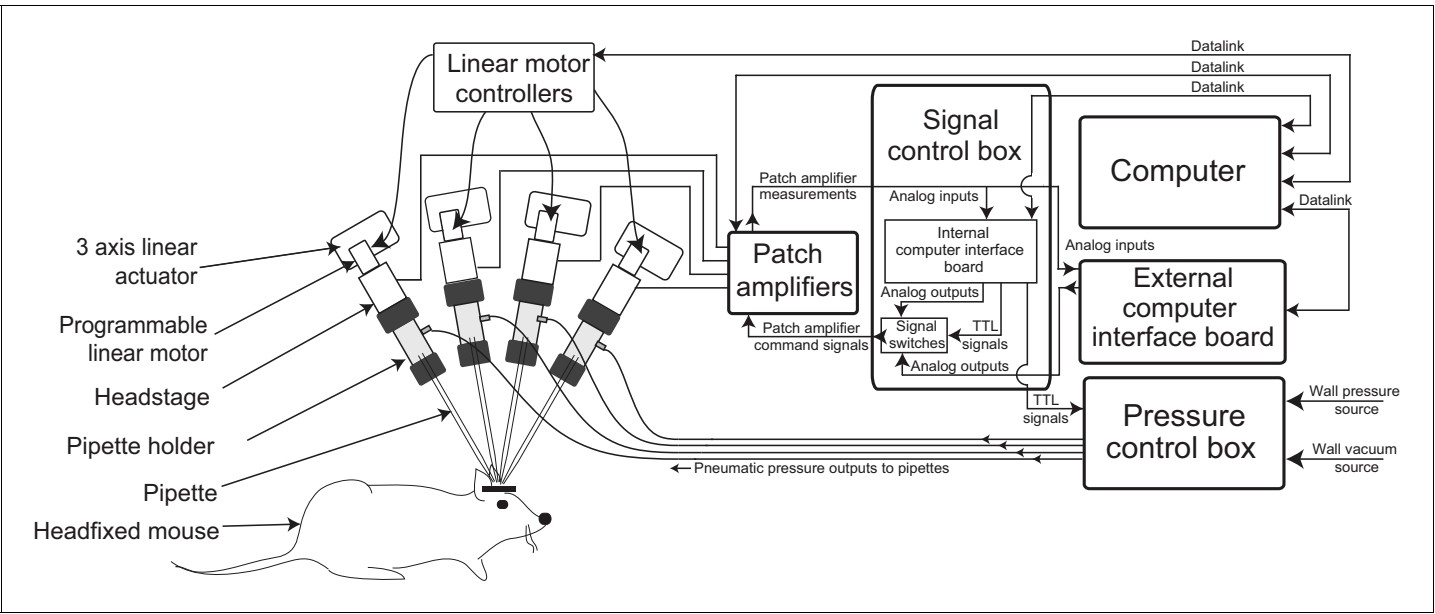

**Figure 1.** Multipatching robot: hardware architecture. (a) Schematic of multipatching robot used for obtaining whole-cell patch recordings from multiple neurons simultaneously in vivo. The system consists of four robotic arms arranged radially and associated signal and pressure control hardware. The pipette, pipette holder and the amplifier headstage are mounted to the robotic arms (see also *Figure 2*). Each headstage is connected to a patch amplifier, which routes signals to a computer via two computer interface boards. A computer interface board, located within the main signal control box, also serves to control the pressure regulation device that can apply pressures ranging from −350 to 1000 mBar independently to each patch pipette (see also *Figure 1—figure supplements 1* and *2*).
DOI: https://doi.org/10.7554/eLife.24656.002

The following figure supplements are available for figure 1:

**Figure supplement 1.** Schematic of the pressure control box.
DOI: https://doi.org/10.7554/eLife.24656.003
**Figure supplement 2.** Schematic of the signal control box.
DOI: https://doi.org/10.7554/eLife.24656.004

signals from the computer interface board and digitizer to the patch amplifiers. Amplified signals from each patch amplifier were sent simultaneously to the analog input channels in both the computer interface board and the digitizer. During multipatcher operation, real time pipette resistance was computed by acquiring the signals from the patch amplifier for use in the multipatching algorithm.

## Pressure control box

The pressures applied to the pipettes during the multipatcher operation—high positive pressure, low positive pressure, low suction pressure, and high suction pressure—were controlled using the pressure control box schematized in *Figure 1—figure supplement 2*. The pressurized air and vacuum inputs are down regulated and switched to control the pressure applied to the pipettes. This enables independent control of the pressures on each pipette. The output of these electronic pressure regulators (VSO-EV series pressure regulators and OEM-PS1 series vacuum regulators, Parker) is controlled by 0–5 V analog voltage signals using potentiometers on the front panel of the pressure control box. These regulators were used to supply the regulated pressure states to the valve bank. Within the 12-valve bank, a set of 3 three-way solenoid valves (LHDA0533215H-A, Lee Company) were dedicated to each pipette. Each valve was switched using transistor-transistor logic (TTL) signals from the digitizer board, connected to the gates of power MOSFETS. This allows the pressure control box to deliver four pressure states independently to each pipette at various stages (i.e., regional pipette localization, neuron hunting, gigasealing, break-in) of multipatcher operation. In addition to the four pressure states enabled by the pressure control box, one additional state,

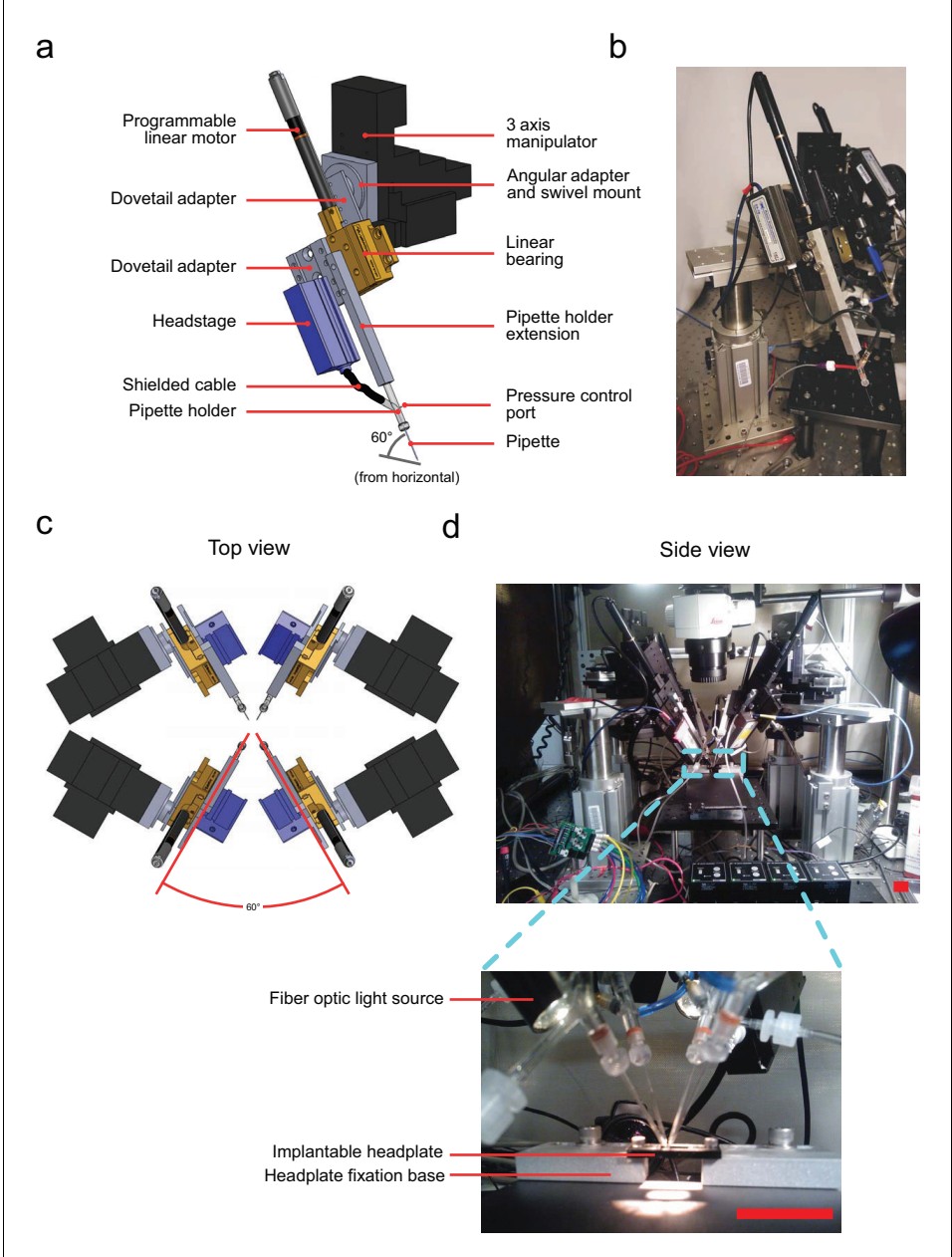

**Figure 2.** Computer aided design (CAD) rendering and photographs of the multipatcher. (a) CAD rendering illustrating details of a single robotic arm that allows 3-axis manual motorized and programmatic control of the pipette's axial position. The pipette is mounted on a pipette holder, which is in turn mounted on a bearing driven by a programmable linear motor. (b) Corresponding photograph showing a single robotic arm. Scale bar indicates 50 mm. (c) Top view rendering of the array of robotic arms illustrating relative arrangement. (d) Side view photograph of the array of robotic arms, with inset showing the arrangement of the pipette tips relative to the head plate affixed to the mouse, the head plate fixation base, and animal warming pad used in anesthetized experiments. Scale bars indicate 25 mm.

DOI: https://doi.org/10.7554/eLife.24656.005

atmospheric pressure, is controlled by a three-way valve located adjacent to the pipette holder for fast switching. The pressure control is instantiated for each pipette independently.

## Multipatcher algorithm

We derived the algorithm for multipatching by iteratively modifying our previously described auto-patching algorithm (*Kodandaramaiah et al., 2012*). We iterated through several strategies for hardware and algorithmic control to maximize scaling, yield, and throughput. However, several steps are common to all the strategies (*Figure 3a*): First, the experimenter installs freshly pulled patch pipettes filled with intracellular pipette solution into all robotic arms, coarsely aligns the pipettes over a single craniotomy, and initiates the computer program used to control the multipatcher robot (time point i, *Figure 3a*). The multipatcher robot then performs an initial assessment of the pipettes' resistances to ensure their resistances are in the acceptable range (3–9 MΩ) (*Kodandaramaiah et al., 2016*). For pipettes that are found to be satisfactory, their positions above the brain surface are noted and all further positions during robot operation are referenced from these initial starting points. The pipettes are then lowered to the desired depths at a speed of ~200 $\mu$m/s while applying high positive pressure. Pipettes in different robot arms can be lowered to different depths, thereby allowing simultaneous recordings, for example, from different layers of the cortex, or even different regions of the brain (time ii, *Figure 3a*). Once lowered to depth, the pressures in the pipettes are decreased to low positive pressure (20–25 mBar) and the pipette resistances are compared to their resistances recorded outside the brain. If resistance increases greater than 0.35 MΩ are detected in any of the pipettes, their tips are deemed blocked or fouled and those pipettes are depressurized to atmospheric pressure and their actuator arms deactivated (time iii, *Figure 3a*). This is analogous to the "regional pipette localization' step in the original autopatcher algorithm (*Kodandaramaiah et al., 2012*).

After regional pipette localization, the first algorithm development strategy was the simplest to implement from a hardware design standpoint: a pressure control box with a single pneumatic valve bank to control pressure state-switching in all the pipettes (*Figure 3b* and *Figure 3—figure supplement 1*). This required an algorithm that accommodated synchronized pressure state switching events in all pipettes. Hence, we first implemented a simple extension of the autopatcher algorithm (*Kodandaramaiah et al., 2012*), as shown in *Figure 3b*. After regional pipette localization (time iii, *Figure 3a*), active pipettes proceeded to neuron hunting. Once a neuron was detected, the corresponding motor was deactivated and the rest of the pipettes continued neuron hunting, until all pipettes encountered neurons. Pressure in all pipettes was simultaneously released and gigasealing was attempted in a manner identical to the autopatcher. In 19 trials (n = 3 mice) where three or more active pipettes performed the neuron hunting and gigasealing tasks, the multipatcher established successful gigaseals 22% of the time (15/68 pipettes, 19 trials; 8/76 pipettes were deactivated at the end of regional pipette localization stage due to tip blockage). The pipettes reaching neurons last, and thereby attempting to establish gigaseal immediately, successfully formed gigaseals 36.8% of the time (7/19 pipettes). In the rest of attempts, successful gigaseals were formed 16.3% of the time (8/49 pipettes). Thus waiting significantly lowered gigaseal yield relative to both not waiting and previously reported gigaseal yields (*van Welie et al., 2016*).

We analyzed the resistance measurements for these "waiting' pipettes and found that in some of them, while waiting for other pipettes, their resistance had fallen to the pipette resistance measured before contact with a neuron (20% of the time, 10/49 trials). This indicates that tissue displacements, caused perhaps by either motion of other pipettes in the brain or the force of the ejected intracellular pipette solution, was large enough to dislodge neurons. Further, only 20.5%, 8 out of the remaining 39 pipettes, established successful gigaseals, even when elevated resistance readings, indicating proximity to a neuron, were observed. We hypothesized that the constant exposure to the intracellular pipette solution while waiting possibly had a deleterious effect on the neurons, resulting in lower rates of gigasealing.

Other issues encountered during this strategy were that the movement of pipette actuator arms during neuron hunting resulted in electrical noise, and when coincident with the resistance measurements in other channels, resulted in spurious readings. Thus, the resistance measurements in all channels needed to be synchronized when pipettes were performing neuron hunting. Also, this approach did not take into consideration brain tissue displacement caused by the motion of multiple pipettes in brain. Since encountering a neuron during blind in vivo patch clamping is a random process, multiple autopatchers running independently encounter neurons at different times. This could cause a problem, for example during gigasealing (*DeWeese, 2007*), when it is critical to prevent any

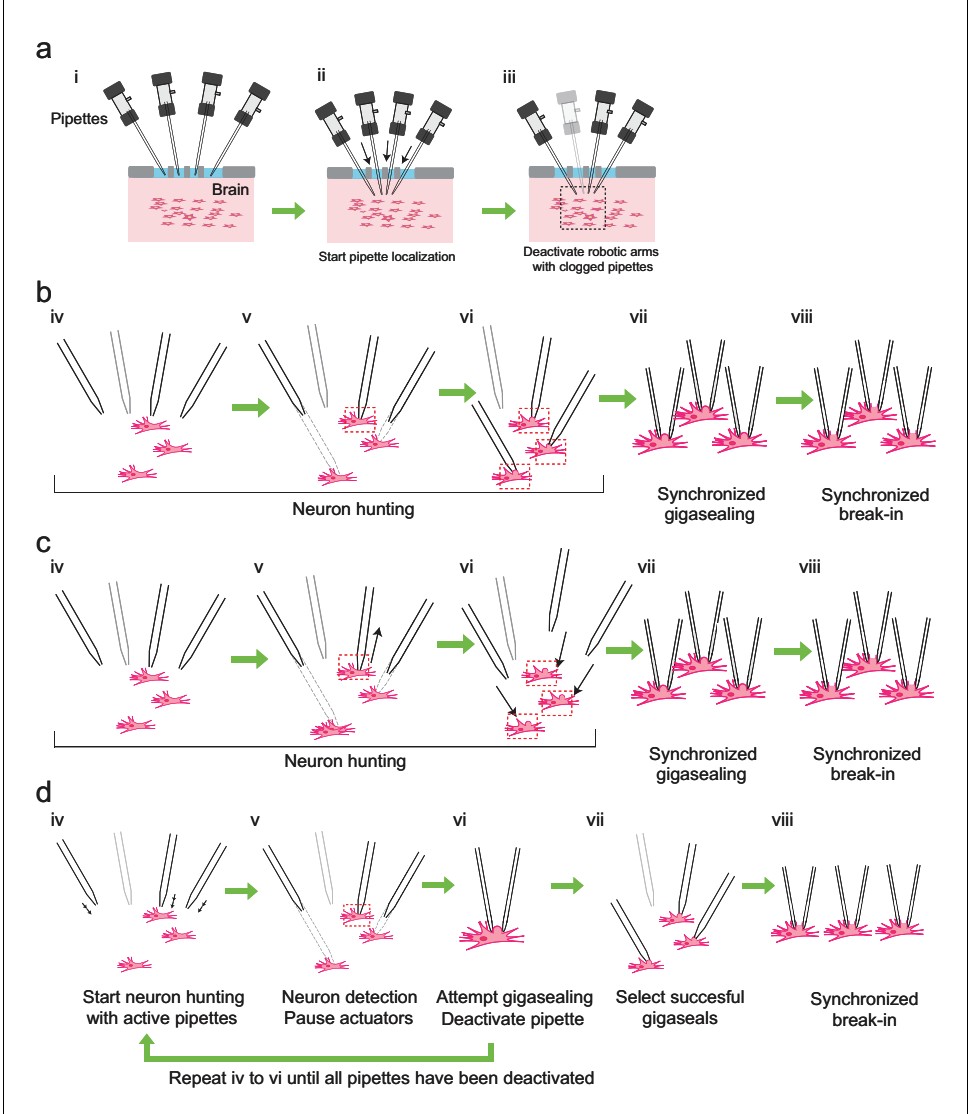

**Figure 3.** Development of the multipatcher algorithm. (**a**) The initial steps of the multipatcher algorithm: (i) The experimenter manually positioned the pipettes in contact with the cortical surface. (ii) The robot automatically lowered all pipettes to the desired target region in the brain. (iii) Clogged pipettes were detected and deactivated (grayed out pipette). The pipettes that were still active continued seeking for neurons. The black box in iii denotes the region zoomed in in the next figure panels. (**b**) Schematic of the first development iteration: (iv) active pipettes continued hunting for neurons. (v) whenever a pipette encountered a neuron (red box) the corresponding motor was deactivated, while the rest continued seeking for neurons. (vi) in this example, all active pipettes have made contact with neurons (red boxes). (vii) gigasealing was attempted simultaneously in all pipettes, by releasing positive pressure, applying suction pressure and applying hyperpolarizing voltages synchronously. (viii) Breaking-in was then attempted synchronously in all cells with successful gigaseals. (**c**) Schematic of the second development iteration: (iv) active pipettes continued hunting for neurons, moving at steps of 2 $\mu$m. (v) each time a pipette encountered a neuron, it was retracted back by 30 $\mu$m (black arrow), and held in that position. (vi) once all pipettes had performed this step, all pipettes were simultaneously lowered down to the positions where they had previously encountered neurons (black arrows). (vii) synchronous gigasealing was attempted. (viii) synchronous break-in was attempted in all gigasealed neurons. (**d**) Third and final development iteration: (iv) The multipatcher moved the pipettes simultaneously at 2 $\mu$m steps (black arrows). (v) when a pipette detected a neuron, all pipettes were halted. (vi) gigasealing was attempted on the single pipette that had detected a neuron. (vii) After the gigasealing procedure was completed, whether successful or not, the remaining pipettes resumed neuron hunting. Steps iv to vi were repeated until gigasealing had been attempted on all active

*Figure 3 continued on next page*

*Figure 3 continued*

pipettes. (vii) pipettes with resistance greater than 1 GΩ were selected by the algorithm to continue to break-in stage. (viii) Break-in was performed simultaneously on all gigasealed neurons.

DOI: https://doi.org/10.7554/eLife.24656.006

The following figure supplement is available for figure 3:

**Figure supplement 1.** Schematics of valve bank's configurations across the different iterations of the algorithm.

DOI: https://doi.org/10.7554/eLife.24656.007

relative motion between the pipette tip and the cell. This highlights the need for a robotic arm interaction strategy: some steps should be synchronous to prioritize throughput, while other steps should be independent to prioritize yield.

We therefore implemented a second algorithm, shown in 3 c. This algorithm proceeded along the same lines as the previous one, until a neuron was encountered at one of the channels, at which time, the pipette was retracted by 30 $\mu$m and stopped. We chose a value of 30 $\mu$m because that was the minimum distance the pipettes needed to be retracted before the resistance measurement decreased to the average baseline value (n = 15 trials). This process was repeated for all the active pipettes, such that at the end of neuron hunting the relative positions of all the pipettes and the corresponding neurons they encountered were the same. As a final neuron-hunting step, all pipettes were moved forward by the same distance (30 $\mu$m), and gigasealing attempted synchronously. This algorithm yielded a success rate for gigasealing of $\sim$20% (12/59 pipettes in 17 trials, with nine pipettes deactivated at the end of regional pipette localization stage due to tip blockage). Again, this was much less than what we would expect when using the autopatcher robot. We analyzed the resistance measurement traces for this algorithm and found that after the final neuron hunting step, when all pipettes advanced forward by 30 $\mu$m, resistances went back to the elevated values indicated by contact with neurons in only 45.7% (27/59 pipettes), likely due to tissue displacement.

The two development iterations described above had synchronous pressure state switching events in all pipettes. This allowed a single pneumatic valve bank (***Figure 3—figure supplement 1a***) to perform the pressure state switching and simultaneously supply the pressure states to all pipettes in parallel. In the third iteration, we decided to perform the gigasealing operation immediately upon detection of a neuron in any one of the pipettes. As it has been observed previously, once either the gigasealed cell attached or whole-cell stage has been achieved, the configuration is remarkably stable against motion artifacts. This has been used previously to record in the whole-cell state from head-fixed rodents (***Poulet and Petersen, 2008***; ***Kodandaramaiah et al., 2012***; ***Kodandaramaiah et al., 2016***; ***Chabrol et al., 2015***; ***Margrie et al., 2002***; ***Margrie et al., 2003***) and freely moving animals (***Epsztein et al., 2010***; ***Lee et al., 2009***; ***Lee et al., 2014***). Several groups have also shown that it is possible to carry out loose cell attached recordings for tens of minutes to hours (***DeWeese et al., 2003***; ***Tang et al., 2014***). Based on these observations, once a pipette encountered a neuron, the program paused neuron hunting in all channels and attempted gigasealing in the channel that encountered a neuron (***Figure 3***).

To implement this third strategy, we re-designed the pressure control box to incorporate a separate pneumatic valve bank for each pipette (***Figure 3—figure supplement 1b***), which allowed independent pressure state switching in each individual pipette. The third and final iteration of the algorithm then proceeded as follows: after regional pipette localization and deactivation of clogged pipettes (***Figure 3a***), the multipatcher robot moves the remaining pipettes in small incremental steps (2 $\mu$m); after each step, square wave voltage pulses (e.g., 10 mV at 10 Hz, with offset voltage set at 0 mV) are applied to the pipettes to compute their resistances (time point iv in ***Figure 3d***). This two-step process of pipette advancement followed by resistance measurement is repeated while looking for monotonic increases of pipette resistance above 0.25 MΩ over three actuation steps taken in one or more patch pipettes that indicate suitable contact with a neuron for patch clamping (time v in ***Figure 3d***), analogous to the 'neuron hunting' stage in the autopatcher operation (***Kodandaramaiah et al., 2012***). After detecting contact with a neuron, the robot halts the movement of all pipettes, and attempts to establish a gigaseal in the pipette that has encountered a neuron (time vi in ***Figure 3d***), analogous to the "gigasealing' stage in the autopatcher (***Kodandaramaiah et al., 2012***) by applying low suction pressure and a hyperpolarizing voltage.

After a gigasealing attempt, lasting 60 secs or less, the pipette's linear motor is deactivated, and the remaining pipettes resume neuron hunting. This alternating neuron hunting and gigasealing cycle is repeated until all pipettes have encountered neurons and attempted to establish gigaseals (time vii in *Figure 3d*). At this point, pipettes that have successfully formed gigaseals are selected and, at the operator's command, the robot applies pulses of suction until it successfully breaks into the gigasealed cells (time viii in *Figure 3d*). Using this algorithm, we were able to get successful gigasealed cell attached states in 58% of the active pipettes (77 out of 133 pipettes in 41 trials, 41 pipettes were deactivated at the end of the initial localization step due to blockage or clogging). This was the highest yield we obtained from all three iterations, and exceeded the gigasealing success rate that we obtained with our autopatcher algorithm (*Kodandaramaiah et al., 2012*). The details of the software implementation and operation of this algorithm can be found in *Supplementary file 4* and Source Code Files 1–4.

## Time course of multipatcher operation

A series of pressure state switching and resistance measurements in four pipettes during a single, typical multipatcher trial in which multiple whole-cell recordings are shown in *Figure 4a and b*. Pressure and resistance measurements from each pipette are coded with green, blue, red, and purple colors. Key events during the trial are denoted by lowercase roman numerals, with the colored bars at the top of *Figure 4b* (*Figure 4—source data 1*) indicating when the robot arms are active, or moving. Grey shaded areas indicate when all robot arms are stopped and a pipette is attempting gigasealing with its contacted neuron. The detection of a neuron in pipette 1 is shown at time i. Between times i and ii, all pipettes paused actuation and gigasealing was attempted with pipette 1 (green). This 60 s gigasealing attempt, started at time i, was conducted as follows: (1) measure resistance at low positive pressure for 10 s to ensure positive confirmation of contact with a neuron, (2) switch to atmospheric pressure for 5 s to allow the cell membrane to begin sealing, (3) apply low suction pressure for 10 s to form gigaseal, (4) return pressure to atmospheric and apply hyperpolarizing voltage of −35 mV, (5) reduce voltage linearly to −70 mV over 30 s, and (6) wait 5 s. The motor of pipette 1 was deactivated for the rest of the trial. At time ii, the remaining pipettes resumed neuron hunting. At time iii the robot detected contact with a neuron using pipette 2. The same gigasealing steps described above for pipette 1 were used for pipette 2 between times iii and iv, resulting in unsuccessful gigaseal formation, as indicated by a minimal resistance increase. The experimenter terminated the gigasealing attempt prematurely, manually, after 35 s, rather than having the robot continue automatically after 60 s as before. Between times v and vi, and again between times vii and viii, the robot successfully gigasealed pipettes 3 and 4. At time ix, the gigasealed neurons attached to the patch electrodes in pipettes 1, 3 and 4 were broken into to establish whole-cell patch recordings. The time for execution of gigasealing tasks for multipatching was fixed at 60 s, whereas in the previous autopatcher, break-in was initiated at the discretion of the operator. The gigasealing times recorded for autopatching were the times taken for gigaseals to fully stabilize and asymptote, upon which break-in was initiated by the operator. In the multipatcher algorithm, however, we used a fixed time for gigasealing with the cell being clamped at −70 mV holding potential at the end of the 60 s gigasealing routine. Thus, even as the program resumed neuron hunting with pipettes that were yet to encounter neurons, the seal resistance continued to increase and asymptote due to the hyperpolarizing holding potential that was applied. This did not however apply to the pipette that attempted gigasealing last for which the usual conditions used for autopatching were applied.

## Performance in anesthetized rodents

We first assessed the performance of the multipatcher robot in anesthetized, head-fixed mice. Representative voltage traces of two and three neurons patched simultaneously in the somatosensory cortex of an anesthetized mouse are shown respectively in *Figure 5a*, right (*Figure 5—source data 1*) and *Figure 5b*, right *Figure 5—source data 2*). Overall in anesthetized animals, for recordings obtained in visual cortex and somatosensory cortex, the multipatcher robot controlling four patch pipettes was able to establish successful whole-cell recordings from multiple neurons in 30.7% (13/41 trials, eight mice) of trials. We were not able to obtain successful recordings from all four pipettes that simultaneously met the quality criteria: resting membrane potential below −50 mV, less than 200 pA of negative current injected to hold the cell at −65 mV in voltage clamp mode, less than 100

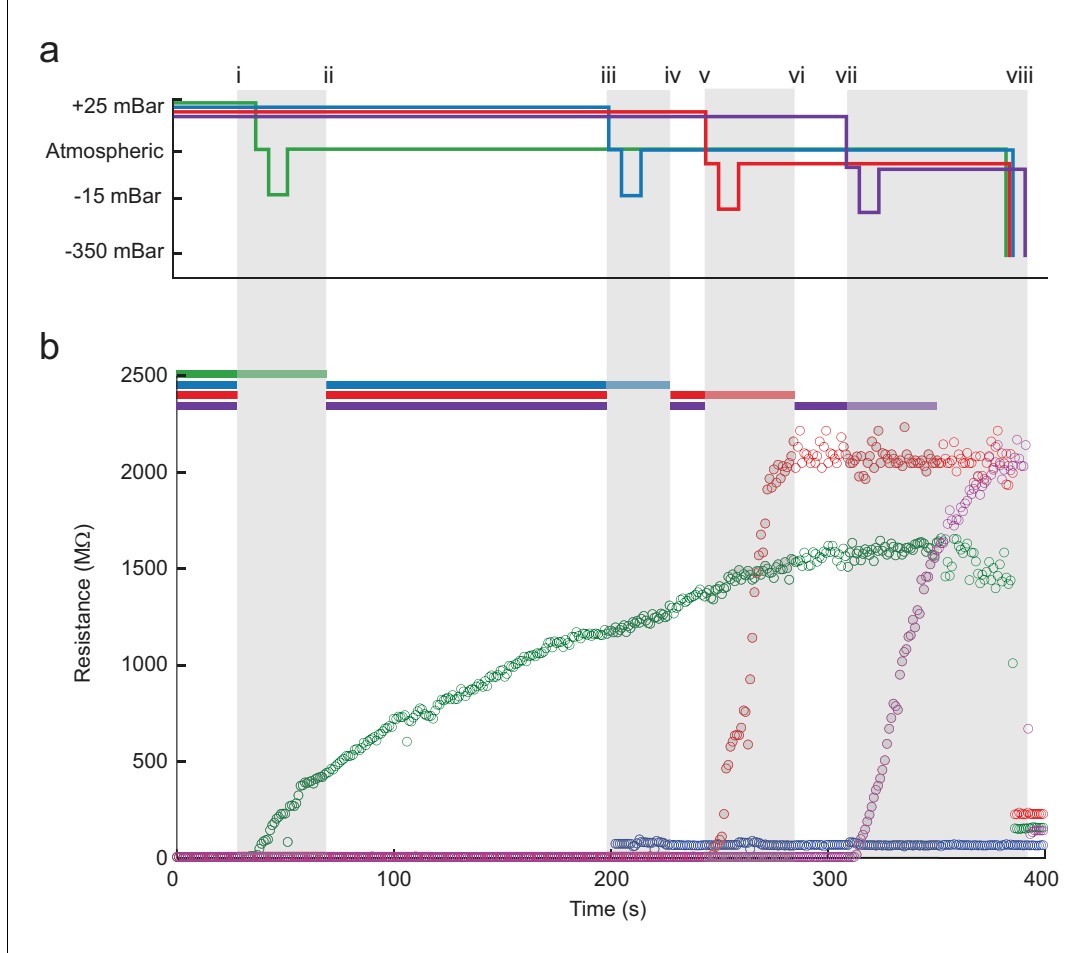

**Figure 4.** The multipatcher robot in operation. (a) Time series of the pressures in each of the four valves during the multipatcher operation. The valve pressure settings, each corresponding to a pipette, are color coded blue, green, red and purple. The roman numerals and gray areas denote different steps of the algorithm operation, as shown in *Figure 3d*. (b) Time series of resistance measurements in a representative multipatching trial. Colors and roman numerals are coded as in (a). The horizontal bars on top indicate the epochs of the algorithm when motors are active and moving, and the grayed out sections indicate epochs when gigasealing was attempted in a pipette. The vertical gray bars indicate epochs in which all pipettes remained stationary. Key events are flagged by roman numerals. Between i and ii gigasealing was attempted in the pipette color-coded green; between iii and iv, gigasealing was attempted in the pipette color coded blue. In this gigasealing attempt, the experimenter utilized a manual override option to prematurely terminate the gigasealing process after observing less than 100 MΩ increase in pipette resistance after 35 s, typically indicative of an unsuccessful gigasealing attempt. Gigasealing was attempted similarly with pipettes color coded red and magenta between time points v to vi and vii to viii respectively. Break-in was attempted in all pipettes that had successfully obtained gigaseals at step ix and obtained whole-cell configurations in all three pipettes.

DOI: https://doi.org/10.7554/eLife.24656.008

The following source data and figure supplement are available for figure 4:

**Source data 1.** Time series of resistance measurements.
DOI: https://doi.org/10.7554/eLife.24656.010

**Figure supplement 1.** Pipette tracks in the somatosensory cortex after a complete experiment.
DOI: https://doi.org/10.7554/eLife.24656.009

MΩ of initial series resistance, and recording time duration of at least 5 min. Recordings were considered to be successful dual- and triple-patches only if all the neurons met the quality and time criteria (*Figure 5c and d*, and *Figure 5—source data 5–7*). Nineteen percent of the pipette tips were blocked or fouled in the initial descent to depth (31/164 pipettes in 41 trials, eight mice), comparable to that obtained previously (*Kodandaramaiah et al., 2012*). Of the pipettes that executed neuron hunting, 57% (77/133 pipettes in 41 trials, eight mice) established successful gigaseals, of which 67.5% (52/77 pipettes in 41 trials, eight mice) yielded successful whole-cell recordings of necessary

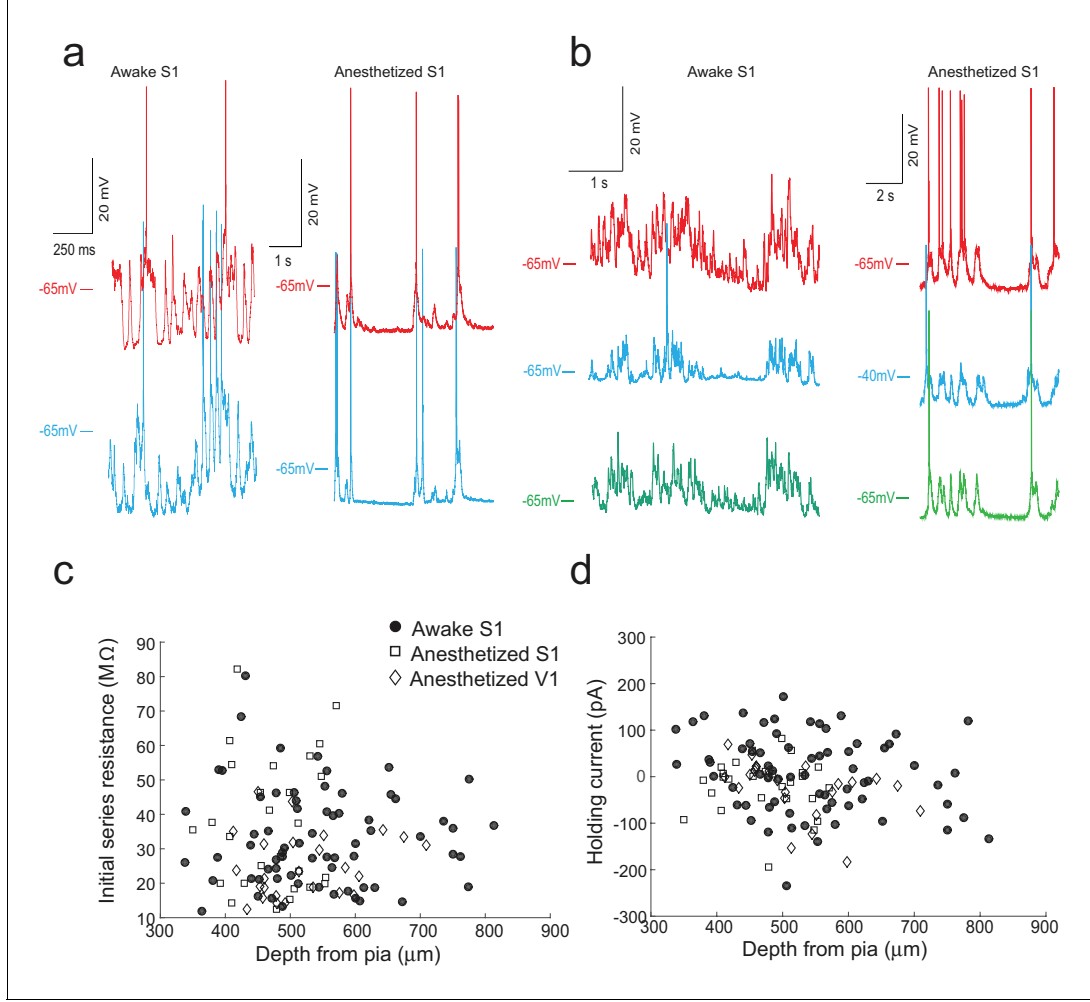

**Figure 5.** The multipatcher robot in operation. (**a**) Representative voltage traces for two neurons patched simultaneously in the somatosensory cortex of awake (left) and anesthetized (right) head restrained mouse. (**b**) Representative voltage traces from three neurons patched simultaneously in the somatosensory cortex of an awake (left) and anesthetized (right) head-fixed mouse. (**c**) Initial series resistances of neurons recorded at various depths in the cortex plotted against the depths from the pia at which the recordings were obtained. (**d**) Holding currents required to hold various whole-cell patched neurons at −65 mV in voltage clamp mode plotted against the depth from the pia at which recordings were obtained.

DOI: https://doi.org/10.7554/eLife.24656.011

The following source data is available for figure 5:

**Source data 1.** Raw traces of a double patch in S1 of an anesthetized mouse.
DOI: https://doi.org/10.7554/eLife.24656.012

**Source data 2.** Raw traces of a triple patch in S1 of an anesthetized mouse.
DOI: https://doi.org/10.7554/eLife.24656.013

**Source data 3.** Raw traces of a double patch in S1 of an awake mouse.
DOI: https://doi.org/10.7554/eLife.24656.014

**Source data 4.** Raw traces of a triple patch in S1 of an awake mouse.
DOI: https://doi.org/10.7554/eLife.24656.015

**Source data 5.** Depth, Series resistance, and holding current during whole-cell recordings in anesthetized S1.
DOI: https://doi.org/10.7554/eLife.24656.016

**Source data 6.** Depth, Series resistance, and holding current during whole-cell recordings in anesthetized V1.
DOI: https://doi.org/10.7554/eLife.24656.017

**Source data 7.** Depth, Series resistance, and holding current during whole-cell recordings in awake S1.
DOI: https://doi.org/10.7554/eLife.24656.018

quality. Overall, each pipette had a 31.7% (52/164 pipettes) chance of establishing a whole-cell recording. We hypothesize that the high percentage of gigaseals obtained can be attributed to improvements made in our surgical procedures, for example using an autodrilling robot (*Pak et al., 2015*) to perform craniotomies.

We did observe a decrease in break-in success rate as compared to our previous study (*Kodandaramaiah et al., 2012*). Of the 77 neurons that were gigasealed, we established successful whole-cell recordings in 52 neurons, achieving a break-in success rate of 67.5%. Multiple reasons could cause this reduction in break-in. Since the algorithm waited until all pipettes had attempted gigasealing, pipettes that encountered neurons first often had to wait a few minutes before break-in was attempted. A small fraction (6.4%, 5/77 gigasealed neurons in 41 trials, eight mice) lost gigaseal attachment to the neurons while waiting. It is also possible that, of the ones that remained giga-sealed, the increased duration of holding at this stage made break-in more difficult, although we did not systematically explore this. Another potential explanation for the decrease in break-in success rate is the design of the pressure controller box. In it, the volume of air in the tubing between the control box and the patch pipettes may be significant since we had to use longer tubing to route pressure to all the pipette actuators. On average, the series resistance of recorded neurons was 31.54 ±16.53 MΩ (n = 52 neurons, eight mice), while holding currents required to clamp the cell in voltage clamp mode at −65 mV were −21 ±54.08 pA (n = 52 neurons, eight mice) (*Figure 6C and D* and *Figure 5—source data 5–7*). The average membrane capacitances and membrane resistances were 66.13 ± 34.8 pF and 99.3 ±44.41 MΩ (n = 52 neurons, eight mice) respectively (*Figure 6A and B* and *Figure 5—source data 5–7*). In the interest of time, we assessed the time duration of stable recordings in only a subset of trials, and found that recordings obtained were of similar duration (52.62 ±9.87 min, n = 18 neurons in 18 trials) to those obtained with the single autopatcher (*Kodandaramaiah et al., 2012*).

## Performance in awake animals

Once validated in anesthetized animals, we attempted to use the multipatcher robot in awake head restrained mice. We optimized our surgical preparation and head restraint protocol to minimize motion of the brain. Representative voltage traces from two and three neurons patched simultaneously in the somatosensory cortex of awake, head-restrained mice are shown respectively in *Figure 5a*, left (*Figure 5—source data 3*) and *Figure 5b* left (*Figure 5—source data 4*). We attempted a total of 97 multipatching trials (with 97 × 4 = 388 pipettes) in 32 awake, head restrained mice, which yielded 70 successful whole-cell recordings that matched the quality criterion (*Figure 5c and d*, and *Figure 5—source data 5–7*). Thus, if each pipette was considered individually, it had an 17.3% chance of obtaining a whole-cell patch recording (67/388 pipettes). In 44.3% (43/97) of the trials, no usable whole-cell patch recordings were obtained, perhaps due to additional motion of the brain as compared to anesthetized mice. At least one whole-cell patch clamp recording was obtained in 55.7% (54/97) of trials, with 17.5% (17/97) of the trials resulting in dual or triple whole-cell patch clamp recordings.

The mean series resistance of recorded neurons was 33.78 ±14.41 MΩ (n = 67 neurons, 32 mice), while the holding current required to keep the cell in voltage clamp mode at −65 mV was 9.81 ±78.94 pA (n = 67 neurons, 32 mice, *Figure 6c and d* and *Figure 5—source data 5–7*). The average membrane capacitance and membrane resistance was 88.18 ± 49.20 pF (n = 67 neurons, 32 mice) and 109.75 ±68.97 MΩ (n = 67 neurons, 32 mice) respectively (*Figure 6c* and *Figure 5—source data 5–7*). Approximately half (48.6%) of recordings obtained in awake head-fixed animals (34/70 neurons, 32 mice), such as those shown in *Figure 5a and b*, were obtained in experiments involving injection of anesthetics for a separate study (data unpublished). In these experiments, ketamine or dexmedetomidine were injected via a cannula implanted in the peritoneal cavity. The injection was typically done after recording in baseline awake state for 4 min, and resulted in a significant increase in motor activity, with 41.2% of the recorded neurons lost as result of this (14/34 neurons, 32 mice). We obtained a motion index from video recordings using a published methodology (*Gao et al., 2014*) and standardized its values across mice using z-scores. The median of the standardized motion index observed during baseline was significantly lower than the median value observed after the injection of the drug ($\tilde{x}_{base}$=−0.39, $P_{25-75}$ = [−0.66–0.12]; $\tilde{x}_{drug}$ = 0.36, $P_{25-75}$ = [−0.23 0.81]; p-value=0.0003, Wilcoxon Signed-Rank test; $\tilde{x}$ denotes the median, and $P_{25-75}$ denotes the 25th and

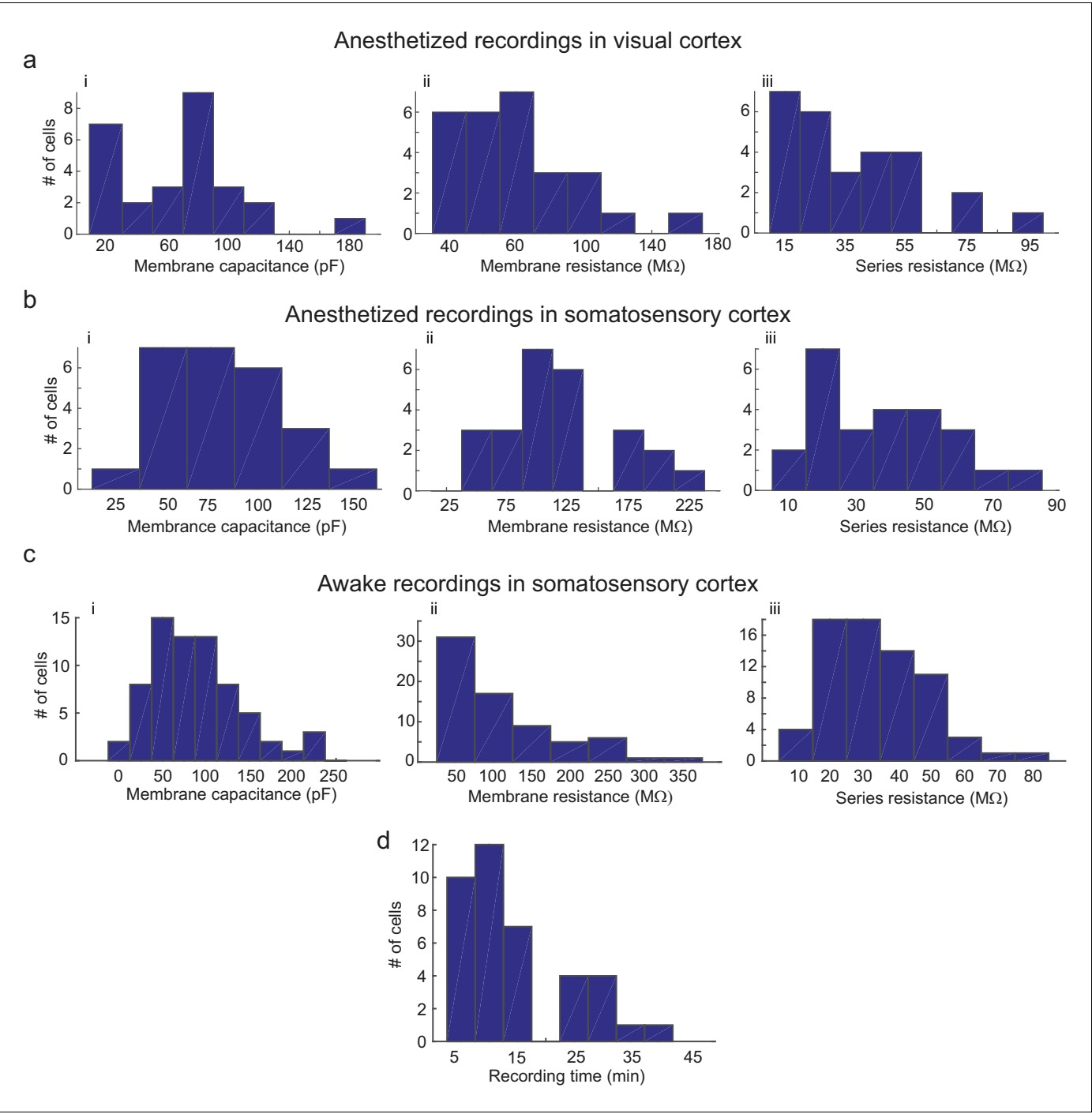

**Figure 6.** Parameters of cells recorded using the multipatcher robot. (**a**) i-iii are histograms of cell membrane capacitance, cell membrane resistance and series resistance, respectively, from neurons recorded in the visual cortex under anesthesia (n = 27 neurons), (**b**) i-iii are histograms of cell membrane capacitance, cell membrane resistance and series resistance, respectively, from neurons recorded in the somatosensory cortex under anesthesia (n = 25 neurons, four mice), (**c**) i-iii are histograms of cell membrane capacitance, cell membrane resistance and initial series resistance, respectively, from neurons recorded in the somatosensory cortex in awake mice (n = 67 neurons). (**d**) Histograms of recording time duration per neuron in awake animals (n = 38 neurons).
DOI: https://doi.org/10.7554/eLife.24656.019

75th percentile values). This analysis provides strong evidence that the period immediately following anesthetic injection does not represent typical motor activity, and therefore we excluded these part of the recording from our calculations of recording duration. After this correction, the remainder of the recordings lasted 13.98 ±10.03 min (n = 36 neurons, *Figure 6d*).

### Throughput and scaling
In a subset of anesthetized trials, we measured the time to manually fill, install and position pipettes for multipatching. In four pipettes the average time taken for filling and installing was 12.5 ±1.4 min (n = 18 trials), and the time for completion of multipatcher trials culminating in successful whole-cell recordings of one of more neurons was 10.5 ±2.6 min (n = 18 trials). Thus, a single pipette requires 3.2 ±0.3 min for installation and 2.6 ±0.6 min for whole-cell patch clamping as compared to the autopatcher (2.0 ± 0.4 for pipette installation and 5 ±2 min for operation). The increased time for pipette installation is due to the increased complexity of tasks involved in positioning with close confinement. However, this is partially offset by the synchronized stages of multipatcher operation, with some additional contribution enabled by limiting gigasealing to 60 s.

In the limit, scaling with this algorithm necessitates that operation time is 70 s/pipette (60 secs for gigasealing and 10 s for assessment). Using these methodologies, we estimate practical limitations to pipette installation are 10–12 pipettes due to hardware constraints, and the time for installation increases non-linearly. Reducing this rate would need a redesign of the actuation modules to enable quick replacement of pipettes for high-throughput operation. Further increases in the throughput may be obtained by using robot systems for automated pipette (*Pak et al., 2011*) and pipette inspection (*Stockslager et al., 2016*), and incorporating pipette cleaning protocols (*Kolb et al., 2016*) between trials to eliminate the pipette changeover time. While the algorithms were formulated and tested using a multipatcher robot controlling four patch pipettes, they can in principle be applied to control many more patch pipettes with further miniaturization. More pipettes would ensure higher a success rate of obtaining multiple patch recordings.

## Discussion
Whole-cell patch clamping is considered a gold standard electrophysiology tool that enables the measurement of both suprathreshold spiking and subthreshold membrane potential fluctuations in single neurons. The ability to whole-cell patch multiple neurons in brain slices has revealed key insights into the mechanisms of synaptic transmission and plasticity (*Perin et al., 2011*). Such circuit level interrogations have been difficult to perform in vivo, particularly in awake animals. Here we demonstrate the ability to use arrays of robotically-guided patch clamping pipettes to simultaneously obtain whole-cell patch clamp recordings in vivo in anesthetized and awake mice. The algorithm used for controlling this robot builds on our previously developed autopatcher algorithm (*Kodandaramaiah et al., 2012*) to control the position and pressure in individual patch pipettes, while taking into consideration the mechanical interactions of pipettes with the brain tissue while seeking neurons in close proximity to each other.

The automation of multi-neuron recording in vivo opens up the possibility of further parallelizing the multipatcher for high density mapping of the intracellular activity of ensembles of neurons in intact tissue, which will be hard to accomplish by humans performing such tasks manually. The advent of low-cost, silicon patch-chip amplifiers (*Harrison et al., 2015*) will enable scaling up the pipette count at a fraction of the cost compared to that of traditional analog amplifiers. Combined with surgical robotics advances that allow precisely defined access to brain (*Pak et al., 2015*), high density robotically guided patch clamping arrays that simultaneously target and record from neurons distributed locally in microcircuits or in multiple brain regions could be developed. Imaging could be used for closed-loop control of multiple pipettes, as has already been done for single patch pipettes (*Suk et al., 2017*; *Annecchino et al., 2017*). Three or four fold increases in the number of simultaneously controlled pipettes could be achieved by using miniaturized micromanipulators, and increasing the number of pneumatic valve banks in the multipatcher control box——both of which are possible using existing off-the-shelf hardware. The macroscopic scale of currently used patch clamping pipettes will limit further increases in number of pipettes targeting the microcircuit. Thus, novel electrode materials, or microfabrication techniques that realize device architectures amenable for in vivo application will have to be developed. The time taken to assemble and position these pipette

arrays precisely at the brain surface will also increase with the number of electrodes. Developing protocols to clean pipette tips for reuse (*Kolb et al., 2016*), improving the yield of regional pipette localization (*Stoy et al., 2017*), or incorporating hardware to robotically exchange pipettes between trials in dense pipette arrays, will be particularly useful to overcome this limitations. However, the users that are less experienced in patch-clamp techniques should always keep in mind that key skills for performing surgery, high quality durectomy, pulling pipettes, making internal solution, controlling pressure lines, etc., still remain a challenge, and time and practice are required to master these skills (*Kodandaramaiah et al., 2016*).

## Materials and methods

**Key resources table** Detail of the key resources needed to build and operate the multipatcher robot (See also **Supplementary files 3** and **4**).

| Reagent type (species) or resource | Designation | Source or reference | Identifiers | Additional information |
|---|---|---|---|---|
| software, algorithm | *Source code 1* | this paper | | Multipatcher control software in LabView Library file format |
| software, algorithm | *Source code 2* | this paper | | Header (.h) file for interfacing with the amplifier. |
| software, algorithm | *Source code 3* | this paper | | Direct link library (.dll) file for interfacing with the amplifier. |
| software, algorithm | *Source code 4* | this paper | | Library (.lib) file for interfacing with the amplifier. |
| other | *Supplementary file 1* | this paper | | Bill of materials to build the multipatcher hardware. |
| other | *Supplementary file 2* | this paper | | Hardware blueprints. |

### Surgical procedures

We conducted all animal work in accordance to federal, state, and local regulations, and following NIH and AAALAC guidelines and standards. The corresponding protocol (#0113-008-16) was approved by the Institutional Committee on Animal Care at the Massachusetts Institute of Technology. Adult male C57BL/6 mice, ~8 weeks old, were purchased from Taconic. During the period before the experiment, the mice were housed in standard cages in the vivarium for at least one week after procurement from the vendor. Food and water were provided *ad libitum* under a 12 hr light-dark cycles. On the day of the experiment, mice were anesthetized using 1–2% isoflurane in pure oxygen and administered buprenorphine (0.1 mg/kg) and meloxicam (1–2 mg/kg) subcutaneously for analgesia. The scalp was shaved and sterilized by scrubbing alternately with betadine solution and 70% ethanol for three times. The eyes were covered with ophthalmic ointment (Puralube, Dechra) and fixed in a stereotaxic apparatus (Kopf Instruments). After making an incision to expose the skull, fascia was removed with a microcurette. Three self-tapping screws (F000CE094, Morris Precision Screws and Parts) were then implanted on the skull, taking care not to thread them into the skull by more than 150 $\mu$m, to ensure that the tip of the screws did not touch the brain surface. A few drops of medical grade cyanoacrylate tissue adhesive (Vetbond, 3M) were applied at the anchor points, suture lines and to attach the skin to the skull as described previously (*Domnisoru et al., 2013*). Then we affixed a custom head plate, made of either stainless steel or delrin, using dental acrylic (C and B Metabond, Parkell).

For animals that underwent anesthetized recordings, the head plate implantation was followed by four craniotomies made in a 2 × 2 grid using an end-mill ($\emptyset$=200 $\mu$m) and an autodrilling robot (*Pak et al., 2015*). To minimize the chance of pipettes colliding during the multipatcher operation, the spacing between the craniotomies was calculated using the relative orientations and distances between the independent robotic arms, and projecting their paths into the brain from the surface, such that they were ~250 $\mu$m apart at the start of the neuron hunting stage. The small diameter of the craniotomies also allowed for minimization of brain motion. *Figure 4—figure supplement 1* illustrates the tracks left by pipettes lowered to 400 micrometers depth from he surface using two opposing arms of the multi patcher. To target layer 4 of the somatosensory cortex, the craniotomies were made at the following coordinates (in mm from bregma): 1 AP, −2.8 ML; 1 AP, −3.2 ML; 1.5 AP, −2.8 ML; 1.5 AP, −3.2 ML. To target layer 4 of the visual cortex, the coordinates were: (in mm from bregma): 2.75 AP, −2.8 ML; 2.75 AP, −3.2 ML; 3.25 AP, −2.8 ML and 3.25 AP, −3.2 ML.

Craniotomies were followed by counter boring the drilled holes using a dental burr ($\emptyset$=500 $\mu$m) to allow easy visualization of the brain surface.

For animals that underwent awake recordings, a thin film of dental acrylic was applied to cover any exposed skull tissue after the head plate fixation, and were then transferred to a warm cage lit by an infrared heating lamp, and kept there until they were fully ambulatory. The health of the mice was carefully monitored, and buprenorphine and meloxicam were administered for analgesia up to 3 days after surgery. They were allowed to fully recover from the surgery for up to one week in the vivarium before we started the behavioral acclimation to the custom restraint setup. On the day of the awake recordings, the mice were anesthetized again using 1–2% isoflurane, fixed in the stereo-taxic instrument, and the thin dental cement layer was carefully removed using a dental burr to re-expose the skull. We inspected the exposed skull surface for signs of inflammation—fluid secretions at the suture lines as well as softened or damp skull tissue. This signs were observed in 10.9% of experiments (5/46 mice) and no recordings were attempted in these subjects. Of those mice that did not show any inflammation, an array of craniotomies was then opened using a similar procedure as described for anesthetized recording sessions above. The skull was then covered with a silicone sealant (Qwik Sil, World Precision Instruments), and animals were head-fixed in the custom restraint setup and allowed to fully recover from anesthesia before attempting recordings, which were per-formed 45–60 min after the opening of the craniotomy.

## Behavioral acclimation

For awake recordings, the animals were affixed in a custom head and body restraint setup, similar to that described by (*Guo et al., 2014*). The mice were allowed to recover for 7 days after surgical implantation of head plates prior to habituation to the restraint setup. Acclimation was carried out for six consecutive days, with training sessions lasting 30, 30, 45, 45, 60 and 60 min on each day. During these training sessions, mice were given undiluted condensed milk at regular intervals as pos-itive reinforcement

## Electrophysiology

The patch pipettes used in the multipatcher robot were pulled from borosilicate glass capillaries (Outer Diameter: 1.2 mm, Inner Diameter: 0.69 mm, Model G120F-4, Warner Instruments) using a standard, filament-based, flaming-brown pipette puller (Model P97, Sutter Instruments) and had resistances between 5–9 M$\Omega$. Pipettes were stored in a closed container and were used within a few hours of fabrication. They were filled with intracellular pipette solution consisting of (in mM): 125 potassium gluconate (with more added to titrate the final solution to $\sim$290 mOsm and a pH of 7.2), 0.1 CaCl$_2$, 0.6 MgCl$_2$, 1 EGTA, 10 HEPES, 4 Mg ATP, 0.4 Na GTP, 8 NaCl. The surface of the brain was kept moist during the multipatching experiment by covering with sterile artificial cerebrospinal fluid (ACSF) consisting of: 135 mM NaCl, 2.5 mM KCl, 10 mM HEPES, 2 mM CaCl$_2$ and 1 mM MgCl$_2$ with pH titrated to 7.3 by addition of NaOH, and osmolarity titrated to $\sim$300 mOsm by adding NaCl (up to 150 mM final concentration).

## Multipatcher robot operation

At the beginning of each multipatcher experiment, patch pipettes were filled with intracellular pipette solution using a thin polyimide/quartz back-filling needle (Microfil, World Precision Instru-ments) and installed in each of the pipette actuators. The solution was filtered using a 0.2 $\mu$m filter (#28145–475, VWR) to minimize internal clogging of the pipettes. We then adjusted the pressure states required during multipatcher operation in the pressure control box, setting the high positive pressure to 800 mBar, the low positive pressure to 25 mBar, the low negative pressure to −15 mBar, and the high negative pressure to −300 mBar. The pipettes were then manually lowered to roughly the center of a corresponding craniotomy in the mouse skull using the three-axis stage. They were gently lowered to make contact with the brain surface (indicated with slight dimpling when visualized in the stereomicroscope) and retracted back until the tips were just above (15–20 $\mu$m) the brain sur-face. Measurements of depths traversed by each of the patch pipettes were referenced from these starting points outside the brain. The multipatcher then performed the steps detailed in the multi-patcher algorithm section. The algorithm was coded and run in Labview 2011 (National Instruments, *Source code 1–4* and *Supplementary file 4*).

The amplifiers were set to voltage clamp mode and square wave command signals were applied simultaneously to all pipettes (10 mV, 10 Hz, via analog outputs on the cDAQ-9174). Amplified signals were sampled at 15 kHz and filtered using a moving average filter (half width, six samples, with triangular envelope). For each pipette, resistance values are computed on line by dividing applied voltage by the peak-to-peak amplitude of the measured current through the pipette. Each resistance value used in the algorithm was the average of five consecutive resistance measurements. During gigasealing and break-in stages (*Figures 3* and *4* and *Figure 4—source data 1*), DC offsets ranging from 0 to −70 mV are applied to the square wave to enhance the formation of gigaseals, as is common practice. To enable accurate measurement of peak-to-peak amplitude during gigasealing, an additional exponential filter (decay rate = 0.001 s) was digitally applied to eliminate stray currents resulting from uncompensated pipette capacitance. After multipatcher operation, whole-cell patched neurons were recorded using Clampex software (Molecular Devices). Signals were acquired at standard rates (e.g., 30–50 KHz), and low-pass filtered (Bessel filter, 10 kHz cutoff). All data was analyzed using Clampfit software (Molecular Devices) and MATLAB (Mathworks).

## Additional precautions for recording in awake mice

We had to take several steps to optimize the surgical and experimental preparation in the awake experiments. In order to minimize brain motion, the animals were placed within a plastic restrainer that prevented movement of the hindhead and subsequent movement of the spinal cord that could be transmitted to the brain via the cerebrospinal fluid. However, this setup allowed unrestrained movement of the forepaws. We also closely inspected the skull surface, at 40x magnification, on the day of the recording for any signs of inflammation, for motion of the skull with respect to the head plate, and for motion of the brain relative to the skull or head plate. In 6.5% of experiments (3/46 mice), we observed relative motion between the head plate and skull when animal exhibited motor activity, possibly due to fatigue experienced by the dental cement used for implantation during repeated acclimation sessions. We did not attempt any awake recording sessions in these animals. In 10.9% of the subjects (5/46 mice), we observed discernible (10–100 $\mu$m) motion of the brain—assessed by observing the relative distance between the brain and edge of the skull at the craniotomy. We did not attempt any automated recordings in these mice. If motion of the skull or brain was not observed through the stereoscope, we would then start the multipatching trial.

## Additional information

### Funding

| Funder | Grant reference number | Author |
|---|---|---|
| New York Stem Cell Foundation | | Edward S Boyden |
| National Institutes of Health | | Gregory L Holst |
| National Science Foundation | | Edward S Boyden |
| National Institutes of Health | R01 EY023173 | Craig R Forest |
| National Institutes of Health | R01-GM104948 | Emery N Brown |
| National Institutes of Health | P01-GM118620 | Emery N Brown |
| Massachusetts General Hospital | | Emery N Brown |
| Picower Institue for Learning and Memory | | Emery N Brown |
| National Institutes of Health | 1R21NS103098-01 | Suhasa B Kodandaramaiah |
| McGovern Institute Neurotechnology Fund | | Suhasa B Kodandaramaiah |

The funders had no role in study design, data collection and interpretation, or the decision to submit the work for publication.

## Author contributions

Suhasa B Kodandaramaiah, Francisco J Flores, Conceptualization, Resources, Data curation, Software, Formal analysis, Validation, Investigation, Visualization, Methodology, Writing—original draft, Writing—review and editing; Gregory L Holst, Conceptualization, Resources, Software, Methodology; Annabelle C Singer, Conceptualization, Resources, Data curation, Formal analysis, Validation; Xue Han, Emery N Brown, Conceptualization, Supervision, Funding acquisition, Writing—review and editing; Edward S Boyden, Craig R Forest, Conceptualization, Resources, Data curation, Software, Formal analysis, Supervision, Funding acquisition, Validation, Investigation, Visualization, Methodology, Writing—original draft, Project administration, Writing—review and editing

## Author ORCIDs

Suhasa B Kodandaramaiah http://orcid.org/0000-0002-7767-2644
Francisco J Flores http://orcid.org/0000-0002-8974-9717
Xue Han http://orcid.org/0000-0003-3896-4609
Craig R Forest http://orcid.org/0000-0001-5343-1769

## Ethics

Animal experimentation: We conducted all animal work in accordance to federal, state, and local regulations, and following NIH and AAALAC guidelines and standards. The corresponding protocol (#0113-008-16) was approved by the Institutional Committee on Animal Care at the Massachusetts Institute of Technology.

## Decision letter and Author response

Decision letter https://doi.org/10.7554/eLife.24656.031
Author response https://doi.org/10.7554/eLife.24656.032

# Additional files

## Supplementary files

• Source code 1. Multipatcher software files. The *Source codes 1–4* include the main multipatcher software library file to be executed in LabView (*Source code 1*), and the accessory files for interfacing with the Multiclamp 700B patch amplifier: a header file (*Source code 2*), a direct link library (*Source code 3*), and a library file (*Source code 4*).
DOI: https://doi.org/10.7554/eLife.24656.021

• Source code 2. Multiclamp Commander Header file.
DOI: https://doi.org/10.7554/eLife.24656.022

• Source code 3. Multiclamp Commander direct link library.
DOI: https://doi.org/10.7554/eLife.24656.023

• Source code 4. Multiclamp Commander library file.
DOI: https://doi.org/10.7554/eLife.24656.024

• Supplementary file 1. Bill of materials for multipatcher hardware. The *Supplementary file 1* contains the bill of materials needed to assemble the multipatcher hardware.
DOI: https://doi.org/10.7554/eLife.24656.025

• Supplementary file 2. Blueprints for multipatcher hardware. The *Supplementary file 2* contains all the CAD files needed to produce the PCB's and custom-made parts of the multipatcher hardware.
DOI: https://doi.org/10.7554/eLife.24656.026

• Supplementary file 3. Multipatcher assembly manual. The *Supplementary file 3* describes all the materials and procedures to assemble the signal and pressure control hardware necessary to operate the multipatcher.
DOI: https://doi.org/10.7554/eLife.24656.027

• Supplementary file 4. Multipatcher software manual. The *Supplementary file 4* describes the installation and use of the multipatcher control software under the LabView system-design platform.
DOI: https://doi.org/10.7554/eLife.24656.028

• Transparent reporting form
DOI: https://doi.org/10.7554/eLife.24656.029

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
