## [Decision Letter]

Thank you for submitting your article "Multi-neuron intracellular recording in vivo via interacting autopatching robots" for consideration by *eLife*. Your article has been reviewed by three peer reviewers, and the evaluation has been overseen by Andrew King as the Senior and Reviewing Editor. The following individuals involved in the review of your submission have agreed to reveal their identity: Randy M Bruno (Reviewer #2); Andreas T Schaefer (Reviewer #3).

The reviewers have discussed the reviews with one another and the Reviewing Editor has drafted this decision to help you prepare a revised submission.

Summary:

This study describes an automated technique for obtaining multiple simultaneous whole-cell recordings in anesthetized and awake, head-fixed mice. Specifically, the authors developed an algorithm for linking 4 autopatching devices in to attempt to obtain simultaneous whole-cell recordings from nearby neurons. They were able to obtain 2 or 3 simultaneously patched neurons 31% of the time in anesthetized animals, and 19% of the time in awake, head-fixed animals. The reviewers agreed that automating whole-cell recording is valuable for improving experimental throughput since this is a labor-intensive procedure, and that this extension of the authors' previous work on automation of patching to several pipettes could have a considerable impact.

Essential revisions:

Although the reviewers considered this paper to be potentially suitable for the extended format of a Technical Report in *eLife*, they agreed that too little information is provided to allow others to assess or reconstruct the 4 channel autopatching system. More details are needed on how to put together this system as well as on the quality of the recordings obtained with it, including a comparison with manual approaches. The paper addresses how to patch additional neurons while minimally disturbing neurons that have already been patched, but does not deal adequately with the issue of how to position pipettes without them running into each other. Furthermore, delicate and/or time-consuming steps are still manually performed or not fully automated. Consequently, while the reviewers thought that this system had considerable potential, they were not yet convinced about the advance it offered over manual patching.

1) The paper needs to provide enough information for people to as easily as possible replicate the construction of the autopatcher system; this includes providing a parts list, software code, drawings, circuits, PCB layouts and setup instructions (perhaps in the form of an appendix).

2) One aspect that remains completely unclear is the alignment of pipettes. With four and potentially in the future more pipettes, there is not only a risk of not knowing exactly where they go but for them to crash into each other. The pipettes are placed on the brain surface manually with apparent freedom of placement in 2-D, and the authors do not mention how the relative positions of the pipettes are tracked during autopatching. Do they have a way to calibrate the relative positions of the tips in 3D as they move towards each other? How are collisions of pipettes prevented? Can this be automated? This is important, otherwise how can experimenters know where to place the pipettes on the brain surface so they don't collide during the automated advance of the pipettes during patching? If alignment procedure is left to chance the user would presumably be limited to targeting cells several 100 μm apart from each other, thus precluding the study of local to medium size circuits. Do the authors have numbers for the actual inter-pipette tip distances for their successful multi-neuron patching attempts to assess how closely they can regularly record multiple neurons?

3) The authors say it took 10.5 min per attempt, but this presumably excludes the manual exchange of pipettes and manual positioning of the pipettes on the brain surface. On lines 162-163 the authors appear to indicate that it would take (3.2 + 2.6)*4 = approx. 23 min between successive 4-pipette patching attempts, assuming no recording time. Also, the manual exchange of pipettes and placement of each of the 4 pipettes on the brain surface using the four 3-axis manual manipulators would seem to be a step that is labor-intensive and requires experimenter care (in the case of brain surface placement), especially because the experimenter would have to ensure that the pipettes don't hit each other during automated patching. Although there is certainly a savings of effort from automating the lowering of the pipettes to search depth, hunting for neurons, sealing, and breaking in, it looks like the experimenter would need to be actively performing operations approximately half of the time, and be present most or all of the time. For multi-pipette patching, the exchange of pipettes and safe placement of all pipettes on the brain surface is significantly more involved than the single-pipette case, and so the relative gain from automating only the patching itself is smaller for multi-pipette compared to single-pipette patching. This is especially true since there does not appear to be an automated anti-pipette-collision procedure.

4) There needs to be an investigation into selection bias – one key advantage of whole-cell patching is low selection bias (i.e. not by firing rate unlike classical unit recordings). However, the thresholds they set e.g. for quality of seal, the pressure when moving pipettes down (and even possibly the shape of the pipette) create a bias in their own right. While this is usually under the user's control, the autopatcher will allow the user to "conveniently forget" about these issues. Thus, this needs to be discussed prominently and augmented by some data on e.g. how different thresholds result in different ratios of interneurons to pyramidal cells.

5) Generally, for evaluating the quality of the recordings there is too little data on stability, R_access as f(t) and very few raw example traces. Showing data as in the original Nature Methods paper but for 3-neuron recordings would give further confidence that superior stability can be achieved. What is the rate of loss of gigaseals that have already been obtained while other pipettes are hunting for neurons and/or attempting a gigaseal?

6) The novelty of the work, beyond the previous manuscripts, lies in obtaining doublets/triplets, but these multi-neuron recordings are poorly described. For example (Introduction, last paragraph) awake recordings are reported to last ~14 minutes on average – is this with regard to individual cells, doublets or triplets? Presumably, this statement is about individual cells, but that is less important here. The awake section of the Results tells us only that 18 out of 97 trials yielded a doublet or triplet. Please tell the reader how many of each, and how many minutes of usable data could be collected from doublets on average and, separately, from triplets on average. Ranges of time would be nice, too. The preceding anesthetized section should also be clear about the frequency and duration of doubles and triplets. These paired recordings need to be held for at least a few minutes for anyone to obtain sufficient data for the simplest experiments. The authors should also show an example of a 3-neuron recording, as they did for 2-neuron traces in Figure 5.

7) The authors should compare and contrast the essential steps and success rate of their multi-neuron patching algorithm with previous manual methods. In particular, Jouhanneau et al., 2015 (cited by the authors) reported that they obtained many examples of 2-4 simultaneously patched nearby neurons in anesthetized animals using two-photon guided targeting. It would be helpful to the field if the numbers of attempts and mice used to obtain the respective datasets could be compared (perhaps via a personal communication from Poulet).

8) In the development section, for the first algorithm they tried, it's not clear why the authors thought it would be possible to attempt to form a gigaseal by releasing suction after a delay (after other pipettes were also in a position to attempt a gigaseal). In the second algorithm they tried, again it is not clear why the authors assumed that pulling back 30 μm before attempting a gigaseal, waiting, then moving by 30 μm again would work. This is probably a strategy that would not work well in standard single pipette patching. After trying fully independent autopatchers (which makes sense to try first), it would seem to have made the most sense to just try the third (and final) algorithm, since it appears to be the same as that used in Jouhanneau et al. The one algorithmic variation that the authors did not report trying, and which could potentially be successful, would be to break in after each gigaseal is formed, instead of simultaneously after all gigaseals were formed.

9) It is important for less experienced users to emphasize that key skills for performing surgery, high quality durectomy, pulling pipettes, making internal solution, controlling pressure lines etc. remain an obstacle – otherwise users will be heavily disappointed if the autopatcher does not permit plug-and-play physiology with the same ease as e.g. photometry.

---

## [Author Response]

Essential revisions:Although the reviewers considered this paper to be potentially suitable for the extended format of a Technical Report in eLife, they agreed that too little information is provided to allow others to assess or reconstruct the 4 channel autopatching system. More details are needed on how to put together this system as well as on the quality of the recordings obtained with it, including a comparison with manual approaches. The paper addresses how to patch additional neurons while minimally disturbing neurons that have already been patched, but does not deal adequately with the issue of how to position pipettes without them running into each other. Furthermore, delicate and/or time-consuming steps are still manually performed or not fully automated. Consequently, while the reviewers thought that this system had considerable potential, they were not yet convinced about the advance it offered over manual patching.

We would like to thank the reviewers for the constructive feedback on the manuscript. We have significantly revised the manuscript, and hope these address the reviewer and editorial concerns adequately.

1) The paper needs to provide enough information for people to as easily as possible replicate the construction of the autopatcher system; this includes providing a parts list, software code, drawings, circuits, PCB layouts and setup instructions (perhaps in the form of an appendix).

We thank the reviewer for the suggestion, and have now included an appendix describing the construction of the multipatcher system, a user manual for the software code, a parts list and associated CAD files.

2) One aspect that remains completely unclear is the alignment of pipettes. With four and potentially in the future more pipettes, there is not only a risk of not knowing exactly where they go but for them to crash into each other. The pipettes are placed on the brain surface manually with apparent freedom of placement in 2-D, and the authors do not mention how the relative positions of the pipettes are tracked during autopatching. Do they have a way to calibrate the relative positions of the tips in 3D as they move towards each other? How are collisions of pipettes prevented? Can this be automated? This is important, otherwise how can experimenters know where to place the pipettes on the brain surface so they don't collide during the automated advance of the pipettes during patching? If alignment procedure is left to chance the user would presumably be limited to targeting cells several 100 μm apart from each other, thus precluding the study of local to medium size circuits. Do the authors have numbers for the actual inter-pipette tip distances for their successful multi-neuron patching attempts to assess how closely they can regularly record multiple neurons?The reviewer raises an interesting point. We have included a section on the alignment of pipettes. In short, to precisely position the pipette relative to each other inside the brain, we utilized the auto-craniotomy robot (Pak et al. 2015) to perform precisely spaced arrays of craniotomies. We have included this information in the Materials and methods section (subsection “Surgical Procedures”, second paragraph), added a Figure supplement to Figure 4 (Figure 4—figure supplement 1).3) The authors say it took 10.5 min per attempt, but this presumably excludes the manual exchange of pipettes and manual positioning of the pipettes on the brain surface. On lines 162-163 the authors appear to indicate that it would take (3.2 + 2.6)*4 = approx. 23 min between successive 4-pipette patching attempts, assuming no recording time. Also, the manual exchange of pipettes and placement of each of the 4 pipettes on the brain surface using the four 3-axis manual manipulators would seem to be a step that is labor-intensive and requires experimenter care (in the case of brain surface placement), especially because the experimenter would have to ensure that the pipettes don't hit each other during automated patching. Although there is certainly a savings of effort from automating the lowering of the pipettes to search depth, hunting for neurons, sealing, and breaking in, it looks like the experimenter would need to be actively performing operations approximately half of the time, and be present most or all of the time. For multi-pipette patching, the exchange of pipettes and safe placement of all pipettes on the brain surface is significantly more involved than the single-pipette case, and so the relative gain from automating only the patching itself is smaller for multi-pipette compared to single-pipette patching. This is especially true since there does not appear to be an automated anti-pipette-collision procedure.

This is indeed a good observation by the reviewer. Our rate limiting step is the exchange and safe positioning of pipettes. While we focused our efforts in this work on automating the procedure for multipatching, we point to recent work done by our groups (Kolb et al. Nat Sci Rep 2016) where we developed pipette cleaning methods that can be used to along with additional programmatically control to realize fully autonomous autopatchers. In this study, we showed that up to 12 trials could be conducted per hour without any human intervention. It is our aim to equip future versions of the multipatcher with such capabilities for fully autonomous operation. Thus, one initial step registering the position of craniotomies and the pipette tips will be sufficient to run the multipatcher robot for long durations of time, eliminating the key bottle neck identified by the reviewer.

4) There needs to be an investigation into selection bias – one key advantage of whole-cell patching is low selection bias (i.e. not by firing rate unlike classical unit recordings). However, the thresholds they set e.g. for quality of seal, the pressure when moving pipettes down (and even possibly the shape of the pipette) create a bias in their own right. While this is usually under the user's control, the autopatcher will allow the user to "conveniently forget" about these issues. Thus, this needs to be discussed prominently and augmented by some data on e.g. how different thresholds result in different ratios of interneurons to pyramidal cells.

We understand the reviewer’s concern about selection bias. In this study we did not perform an analysis of selection bias because we did perform a detailed analysis in our original Autopatcher paper (Kodandaramaiah et al., 2012 Nat. Methods), and each pipette in the Multipatcher is by itself executing the original Autopatcher algorithm, with all parameters conserved. However, based on our and other studies (e.g.: Rancz et al., 2011 Nat Neurosci), we expected to see about 2/3 of all the neurons recorded to be regular spiking neurons.

5) Generally, for evaluating the quality of the recordings there is too little data on stability, R_access as f(t) and very few raw example traces. Showing data as in the original Nature Methods paper but for 3-neuron recordings would give further confidence that superior stability can be achieved. What is the rate of loss of gigaseals that have already been obtained while other pipettes are hunting for neurons and/or attempting a gigaseal?

We thank the reviewer for the suggestion. We do report the rate of loss of gigaseals when other pipettes are hunting in the last paragraph of the subsection “Multipatcher Algorithm”. We have also added an example raw trace of triple whole-cell patch clamp recording in both anesthetized and awake mice in Figure 5.

6) The novelty of the work, beyond the previous manuscripts, lies in obtaining doublets/triplets, but these multi-neuron recordings are poorly described. For example (Introduction, last paragraph) awake recordings are reported to last ~14 minutes on average – is this with regard to individual cells, doublets or triplets? Presumably, this statement is about individual cells, but that is less important here. The awake section of the Results tells us only that 18 out of 97 trials yielded a doublet or triplet. Please tell the reader how many of each, and how many minutes of usable data could be collected from doublets on average and, separately, from triplets on average. Ranges of time would be nice, too. The preceding anesthetized section should also be clear about the frequency and duration of doubles and triplets. These paired recordings need to be held for at least a few minutes for anyone to obtain sufficient data for the simplest experiments. The authors should also show an example of a 3-neuron recording, as they did for 2-neuron traces in Figure 5.

We have now included representative triple patch recordings obtained in anesthetized and awake head-fixed conditions in Figure 5 of the revised manuscript. Regarding the times, as described in the manuscript, successful recordings were those that met the quality criterion for >5 minutes. This includes useable recordings, as well as the time it took to transition to the recording software, measure the cell parameters. Recordings were considered to be successful duals and triples only if all the neurons met the quality and time criterion. We have included this information in the revised manuscript.

7) The authors should compare and contrast the essential steps and success rate of their multi-neuron patching algorithm with previous manual methods. In particular, Jouhanneau et al., 2015 (cited by the authors) reported that they obtained many examples of 2-4 simultaneously patched nearby neurons in anesthetized animals using two-photon guided targeting. It would be helpful to the field if the numbers of attempts and mice used to obtain the respective datasets could be compared (perhaps via a personal communication from Poulet).

We agree with the reviewer that the field of patch clamping would benefit from a comparison of yields of manual, image guided, and automated procedures. To investigate the yields that manual multi-patch clampers expect, we reached out to the authors of the Jouhanneau et al. 2015 paper but they did not respond.

In our experience, very few articles report detailed step-by-step yields that would enable the aforementioned. We note that two photon targeted patch clamping (TPTP), the technique employed by Jouhanneau et al. 2015, typically has higher reported success rates than blind in vivo patch clamping. Jouhanneau et al. 2015 neither report success rates for individual patch clamp trials, nor do they report a combinatorial reduction in whole-cell yield. They use the shadow patching technique pioneered by Kitamura et al. in 2008. In this paper, Kitamura et al. note a success rate of single patch attempts as 67-70%. Because this technique is an image guided technique, we take this to be the upper bound of yield for patch clamping techniques. Thus, we believe that a 31% yield for any given pipette is reasonable for an automated, blind patch clamping technique.8) In the development section, for the first algorithm they tried, it's not clear why the authors thought it would be possible to attempt to form a gigaseal by releasing suction after a delay (after other pipettes were also in a position to attempt a gigaseal). In the second algorithm they tried, again it is not clear why the authors assumed that pulling back 30 μm before attempting a gigaseal, waiting, then moving by 30 μm again would work. This is probably a strategy that would not work well in standard single pipette patching. After trying fully independent autopatchers (which makes sense to try first), it would seem to have made the most sense to just try the third (and final) algorithm, since it appears to be the same as that used in Jouhanneau et al. The one algorithmic variation that the authors did not report trying, and which could potentially be successful, would be to break in after each gigaseal is formed, instead of simultaneously after all gigaseals were formed.

The reviewer raises a good point. We did not attempt the break-in immediately after gigasealing, as (i) we wanted to revert at the earliest to searching for neurons in the remaining channels and (ii) once whole cell is established, the cell starts getting dialyzed and the access resistances start degrading. Thus, the recording obtained from different pipettes in the same sessions would have been qualitatively different. We did observe that this lead to decreased success rate of breaking in, which is discussed in the manuscript (subsection “Performance in Anesthetized Rodents”, last paragraph).

9) It is important for less experienced users to emphasize that key skills for performing surgery, high quality durectomy, pulling pipettes, making internal solution, controlling pressure lines etc. remain an obstacle – otherwise users will be heavily disappointed if the autopatcher does not permit plug-and-play physiology with the same ease as e.g. photometry.We do agree with the reviewer’s comment. We have edited our Discussion section (last paragraph) to emphasize this point, and added a citation to our previous protocols paper (Kodandaramaiah et al. Nat Protocols 2016) that discusses these nuances in detail.